# Cumulative Dose of Prostaglandin E1 Determines Gastrointestinal Adverse Effects in Term and Near-Term Neonates Awaiting Cardiac Surgery: A Retrospective Cohort Study

**DOI:** 10.3390/children10091572

**Published:** 2023-09-19

**Authors:** Noa Ofek Shlomai, Gilad Lazarovitz, Benjamin Koplewitz, Smadar Eventov Friedman

**Affiliations:** 1Department of Neonatology, Hadassah Medical Center, Faculty of Medicine, Hebrew University of Jerusalem, Jerusalem 91120, Israel; noaofek@hadassah.org.il (N.O.S.); gilad@hadassah.org.il (G.L.); 2Department of Radiology, Hadassah Medical Center, Faculty of Medicine, Hebrew University of Jerusalem, Jerusalem 91120, Israel; benjamink@hadassah.org.il

**Keywords:** neonates, ductal-dependent disease, congenital cardiac defect, prostaglandin E1

## Abstract

Objective: This study aimed to assess the association between treatment characteristics of prostaglandin E1 including initiation time and duration, maximal and cumulative doses, and adverse effects. Design: A retrospective cohort study in which medical records of neonates with duct-dependent lesions were studied for treatment parameters and adverse effects. Multivariable logistic regression model was applied for testing the effect PGE1 variables on outcomes. Main outcome measures: The primary outcomes of this study were association of adverse effects of PGE1 treatment with maximal dose, cumulative dose, and treatment duration. The secondary outcomes included safety of feeding in infants treated with PGE1. Results: Eighty-two infants with duct-dependent lesions receiving PGE1 were included. Several infants who received early PGE1 treatment required ventilation support. Feeds were ceased more often as the cumulative dose and duration of PGE1 treatment increased. Gastrointestinal adverse effects were significantly associated with the cumulative dose of PGE1 and treatment duration. Apneas, hyperthermia, and tachycardia were associated with maximal dose. Our data did not demonstrate a difference in the incidence of NEC associated with characteristics of PGE1 treatment. Conclusion: Cumulative PGE1 dose is associated with gastrointestinal adverse effects in neonates. Lower doses should be considered in neonates expecting prolonged PGE1 treatment.

## 1. Introduction

During fetal life, the ductus arteriosus connects the pulmonary artery to the descending aorta, enabling blood from the right ventricle to bypass the non-aerated lungs directly to the aorta. After birth, increased oxygen saturation and decreased endogenous prostaglandins promote ductal closure. Although the incidence of congenital heart defects is quite high (0.8–1% of all infants), ductal-dependent lesions are relatively rare, estimated in different studies at around 12.5/10,000 live births [1,2].

In infants with severe restriction of blood flow to the pulmonary arteries or the aorta due to congenital cardiovascular defects, duct patency is critical for survival while awaiting surgical repair [3]. The ductus arteriosus is derived from the distal portion of the embryonic sixth brachial arch and is approximately the diameter of the descending aorta in utero. In a normal term infant, the ductus arteriosus constricts soon after birth, stimulated by the rapid postnatal increase in arterial oxygen tension. The muscularis of the ductus is uniquely responsive to oxygen, reacting to an increase in ambient oxygen with sustained contraction. The mechanisms of this response are not fully understood, but the cytochrome P450 system (CYP450), endothelin-1 (ET-1), and intracellular oxidation–reduction (redox) balance appear to be important. In healthy term neonates, glucocorticoids may have an essential role in maturation of the ductal oxygen-sensing apparatus, as antenatal exposure to glucocorticoids is associated with both increased expression of genes for calcium and potassium channels implicated in the oxygen response and decreased risk of persistent ductal patency. This direct response is augmented by a rapid decline in circulating levels of PGE2, a potent ductal smooth muscle relaxant that has a major role in keeping the ductus arteriosus open in the fetus. The placenta is the primary source of fetal PGE2, leading to a precipitous fall in circulating levels upon umbilical cord clamping. Maximal effects of PGE2 withdrawal seem to require antenatal priming of the ductal muscle by the rising levels of PGE2 normally seen late in gestation. Dynamic functional closure initiates at the pulmonary end of the ductus and usually is complete within the first 4 days after birth, but anatomic obliteration is not achieved until after 1 week of age [4].

Prostaglandin E1 (PGE1), or its synthetic form, Alprostadil, may serve as a treatment for relaxing the ductus arteriosus smooth muscle so that the ductus remains patent in neonates with congenital cyanotic cardiac defects [5]. Hiriashi et al., in a study using cross-sectional and Doppler echocardiography, have demonstrated the efficacy of PGE1 treatment in infants with prominent narrowing of the duct, but not when the duct was closed or wide open [6]. PGE1 is administrated as a continuous intravenous infusion. Initial dose is 0.05–0.1 mcg/kg/min, while maintenance and maximum doses are 0.01 and 0.4 mcg/kg/min, respectively.

We have identified two case reports of oral PGE1 administration in newborns with duct-dependent congenital heart defects. PGE1 was given at a dose of 1.5–2 mcg/kg/day. One was in an infant with trisomy18 and hypoplastic left heart syndrome [7]. The other was a small for gestational age newborn with double outlet right ventricle and severe pulmonic stenosis. In both cases, the authors reported that the oral derivative of PGE1 was as effective as the intravenous PGE1 in maintaining ductal patency [8]. To the best of our knowledge, oral PGE1 is not available in Israel, and is not routinely used worldwide.

A recent Cochrane review found no randomized controlled trials designed to determine the safety and efficacy of PGE1. They reviewed open comparative studies of newer formulations of PGE1 and indicated different dosing protocols [9].

When PGE1 treatment is initiated early in life, due to antenatal diagnosis, or early presentation, and infants still have an open ductus arteriosus, lower doses of PGE1 of 0.005–0.01 mcg/kg/min may be sufficient to maintain ductal patency. In older infants, much higher doses of up to 0.1 mcg/kg/min may be necessary for ductal patency [10]. The recommended initial dose of PGE1 used to maintain the ductus arteriosus patent is between 0.05–0.1 mcg/kg/min [11]. However, a much lower dose of 0.01–0.03 mcg/kg/min may be adequate for maintaining patency of the ductus arteriosus with sufficient flow, with fewer complications such as apnea, fever, and hypotension [12].

A study of 154 infants with duct-dependent lesions found that PGE1 dose of 0.01 mcg/kg/min was efficient in 83% infants. Furthermore, infants with systemic obstruction were more likely to require a higher dose [13].

PGE1 treatment may be accompanied with multiple adverse effects. Common adverse effects of intravenous administration of PGE1 include flushing, apnea, and fever, occurring in more than 10% of the cases. Less common adverse effects include bradycardia, tachycardia, edema, hypertension or hypotension, seizure, hypokalemia, diarrhea, disseminated intravascular coagulation, sepsis, gastroesophageal reflux disease, hyperbilirubinemia, jitteriness, hyperemia, irritability, thrombocytopenia, hypoglycemia, peritonitis, and more [5]. Multiple case reports describe neonates who developed periostitis following prolonged PGE1 treatment. In these cases, the periostitis has regressed gradually following discontinuation of PGE1 [14,15].

In addition, prolonged PGE1, for over a week, has been associated with increasing pyloric wall thickening and a “pyloric stenosis like” clinical picture [16].

A single case report describes neonatal heart block associated with low-dose prolonged PGE1 treatment [17].

Aykanet et al., in a study of 35 neonates with duct-dependent lesions receiving PGE1, have recorded multiple adverse effects, but have not found an association with starting, maximal, or cumulative doses of PGE1 [18].

Our study aimed to examine different characteristics of PGE1 treatment (time of initiation, maximal dose, cumulative dose, and treatment duration) and to correlate them with PGE1 adverse effects.

## 2. Methods

### 2.1. Study Population

Full- and near-term term newborns diagnosed with a duct-dependent congenital heart defect in a single tertiary center born between 2011–2016 who were treated with PGE1 were selected. Exclusion criteria included additional major congenital and chromosomal abnormalities, and infants who succumbed before completing 96 h of age.

The study was conducted in accordance with the Declaration of Helsinki, and the protocol was approved by the Ethics Committee of Hadassah Medical Organization ethical committee # HMO-0153-016 15.3.2016.

### 2.2. Prostaglandin E1 Formulation

In our unit, we use **PROSTIN VR** PEDIATRIC Sterile Solution, 0.5 mg/mL, by Pfizer.

### 2.3. Data Collection

This is a retrospective cohort study. Demographic and clinical data were collected from medical records. Data of adverse effects of PGE1 treatment, maximal, daily and cumulative dose (calculated per body weight), and treatment duration (days) were documented. All available radiographs and sonographic examinations were reviewed by a pediatric radiologist, searching for evidence of necrotizing enterocolitis, bone cortical hyperostosis, gastric antral mucosal hypertrophy detected by abdominal ultrasound, or nephrocalcinosis.

### 2.4. Adverse Effects of PGE1

In order to simplify analysis, adverse effects were divided into groups as follows:Gastrointestinal: feeding intolerance (gastrointestinal reflux, vomiting, and residues over 25% of feed), number of feeding cessations during the infants’ hospital stay, necrotizing enterocolitis (NEC, stage 2 and over according to Bell’s criteria [19]), diarrhea, and gastric antral mucosal hypertrophy.Cardiovascular: edema, flushing, heart failure, arrhythmia, bradycardia < 100 bpm, tachycardia > 180 bpm, and hypotension requiring medical treatment.Miscellaneous: hyperthermia (defined as body temperature > 37.5 °C, apnea, bleeding, seizure, jitteriness.

### 2.5. Outcomes

The primary outcomes of this study were the association of short-term adverse effects of PGE1 treatment with maximal dose, cumulative dose, and treatment duration.

The secondary outcomes were the safety of feeding in infants treated with PGE1, assessed by feeding intolerance and development of NEC.

### 2.6. Statistical Analysis

The comparison of a quantitative variable between two independent groups was performed by using either the two-sample *t*-test or the non-parametric Mann–Whitney test. The comparison of quantitative variables between three independent groups was carried out by applying the Kruskal–Wallis non-parametric test. The Pearson correlation coefficient and the non-parametric Spearman correlation coefficient were calculated for assessing the strength of the association between two quantitative variables. Non-parametric tests were used due to small sample sizes and the non-normal distribution of some of the variables assessed. The multivariable logistic regression model was applied for testing the effect PGE1 variables on a dichotomous-dependent outcome variable, correcting for the effect of background variables.

All tests applied were two-tailed, and a *p*-value of 0.05 or less was considered statistically significant.

## 3. Results

Eighty-two newborn infants with ductal-dependent congenital heart defects were included in this study. General characteristics of the study group and PGE1 treatment doses are displayed in Table 1. PGE1 treatment was initiated during the first seven days of life, with a median dose of 0.05 mcg/kg/min, and a cumulative median PGE1 dose of 1035 (99–10,328) mcg/kg. PGE1 treatment was continued for a median of 5 days (Table 1). Fifty infants (61%) required endotracheal ventilation during their preoperational treatment. Congenital heart defects included left, right, and mixed flow obstructions, and most were diagnosed prenatally (Table 2).

The most common adverse effect of PGE1 treatment was apnea (20.7%), followed by hyperthermia (18.3%), edema (18.3%), and gastrointestinal symptoms (13.4%). Fifty-nine percent of the infants had a central line placed, and seven infants had sepsis (8.5%). Adverse effects of PGE1 in our group of infants are detailed in Table 3.

We examined the association of different variables of PGE1 treatment, including age in days at the beginning of PGE1 treatment, maximal and cumulative PGE1 dose, and the duration of treatment, with the incidence of various adverse effects. We found that significantly more infants who received early PGE1 treatment required ventilation support, and there was a trend towards more ventilated infants as the maximal dose of PGE1 increased. In addition, our data show that feeds were ceased more often as the cumulative dose and duration of PGE1 treatment increased (Figure 1). Gastrointestinal adverse effects were significantly associated with the cumulative dose of PGE1 and treatment duration (Figure 2). A receiver operating characteristic (ROC) curve of cumulative dose and gastrointestinal adverse effects demonstrated an area under the curve value of 0.715, with a negative predictive value of 94% for cumulative doses of less than 1352 mcg PGE1 and positive predictive value of 23.6% for larger doses.(Figure 3). Our data did not demonstrate a difference in the incidence of NEC, pneumatosis intestinalis, or the presence of portal air associated with characteristics of PGE1 treatment. Hyperthermia and apnea were borderline (*p* = 0.052) associated with maximal, but not cumulative, dose of PGE1. Regarding cardiovascular adverse effects, we found tachycardia was associated with maximal PGE1 dose, and peripheral edema was associated with the duration of PGE1 treatment with only a trend toward an association with PGE1 cumulative dose.

Our results are summarized in Table 4.

**Figure 1 children-10-01572-f001:**
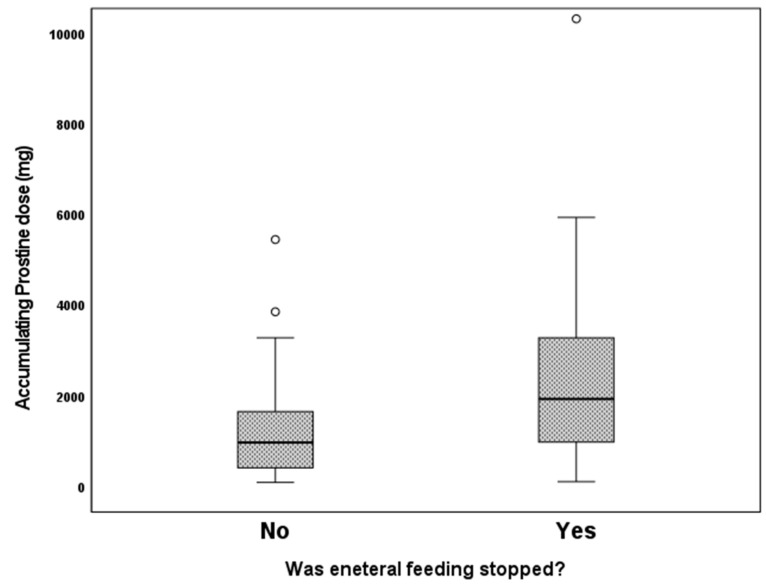
Feeding cessation according to cumulative PGE1 doses. *p* = 0.002.

**Figure 2 children-10-01572-f002:**
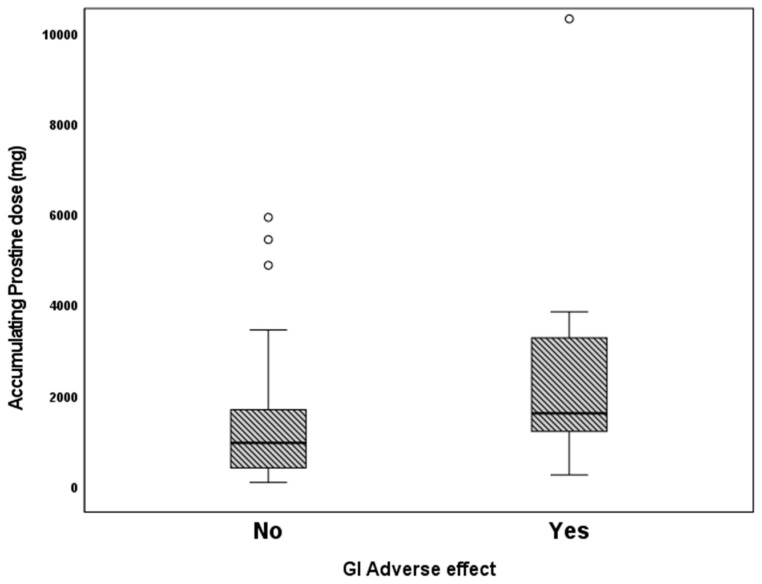
Gastrointestinal adverse effects vs. according to cumulative PGE1 doses. *p* = 0.02.

**Table 4 children-10-01572-t004:** Association of PGE1 dose and duration of treatment with adverse effects.

Adverse Outcome *	Age at Beginning of PGE1 (Days) Median [min–max]	Maximal PGE1 Dose (mcg/kg/min) Median [min–max]	Cumulative PGE1 Dose (mfg/kg) Median [min–max]	Duration of Treatment (Days) Median [min–max]
Ventilation Yes No	0(0–0) 0(0–7) *p* = 0.024	0.05(0.03–0.1) 0.05(0.03–0.1) *p* = 0.090	1213.2(116.3–10,328) 1006.2(99–5454) *p* = 0.987	5.0(0.5–39) 5.0(1–25) *p* = 0.739
Apnea Yes No	0(0–0) 0(0–7) *p* = 0.207	0.05(0.05–0.1) 0.05(0.03–0.1) *p* = 0.052	867(116.3–10,328) 1.044(99–4886.5) *p* = 0.962	4.5(0.5–39) 5(1–36) *p* = 0.733
Enteral feeds Yes were stopped No	0(0–0) 0(0–7) *p* = 0.161	0.05(0.02–0.1) 0.05(0.03–0.1) *p* = 0.329	1938(116.3–10,328) 978.3(99–5454) *p* = 0.015	9(0.5–39) 5(1–25) 0.024
GI adverse Yes effects No	0(0–0) 0(0–7) *p* = 0.309	0.05(0.05–0.1) 0.05(0.3–0.1) *p* = 0.564	1622(267.4–10,328) 970.2(99–5944) *p* = 0.029	5.5(1–39) 5(0.5–36) *p* = 0.451
Hyperthermia Yes No	0(0–0) 0(0–7) *p* = 0.207	0.05(0.05–0.1) 0.05(0.03–0.1) *p* = 0.052	1213.2(174.2–10,328) 1030.8(99–5454) 0.962	4(1–39) 5(0.5–36) *p* = 0.89
Edema Yes No	0(0–1) 0(0–7) *p* = 0.716	0.05(0.03–0.1) 0.05(0.03–0.1) *p* = 0.755	1385.5(180.3–10,328) 970.2(99–3859.7) *p* = 0.072	7.5(2–39) 4(0.5–15) *p* = 0.003
Bradycardia Yes No	0(0–2) 0(0–7) *p* = 0.148	0.05(0.05–0.05) 0.05(0.03–0.1) *p* = 0.856	806.4(560.6–1108.8) 1039.8(99–10,328) *p* = 0.617	4(3–5) 5(0.5–39) *p* = 0.588
Tachycardia Yes No	0(0–0) 0(0–7) *p* = 0.5764	0.1(0.05–0.1) 0.05(0.03–0.1) *p* = 0.006	5943.6(228.2–10,328) 1030.8(99–5454) *p* = 0.236	23(1–39) 5(0.5–36) *p* = 0.284
Hypotension Yes No	0(0–0) 0(0–7) *p* = 0.574	0.1(0.05–0.1) 0.05(0.03–0.1) *p* = 0.856	267.4(254–1401.2) 1039.8(99–10,328) *p* = 0.247	1(1–11) 5(0.5–39) *p* = 0.361

* All adverse outcomes are represented as yes/no.

**Figure 3 children-10-01572-f003:**
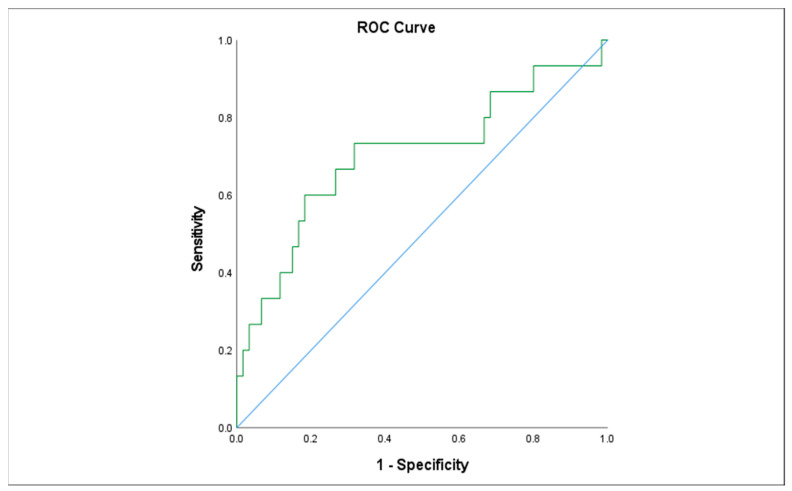
Receiver operating characteristic (ROC) curve describing the association between cumulative PGE1dose and gastrointestinal adverse effects.

## 4. Discussion

In our study of 82 newborns with ductal-dependent congenital heart defect, we evaluated the association between time of initiation, maximal and cumulative dose of PGE1, and the duration of treatment with adverse effects. We report an association between cumulative PGE1 dose and duration of treatment with interruption of enteral feeds and gastrointestinal adverse effects. This information is particularly important when considering pre-operative treatment and planning the optimal timing of surgical repair.

In a study of 35 newborns who received PGE1 for ductal patency due to a congenital heart defect, Aykanat et al. reported 11 of the newborns had experienced 18 side effects of the treatment, including electrolyte imbalance, metabolic alkalosis, gastric outlet obstruction, apnea, and increased urine output. Of these infants, most experienced only one side effect of the treatment. Interestingly, five of their patients experienced gastric outlet obstruction after a week (two patients) and two weeks (three weeks). They did not find a statistically significant association between the initial dose, minimal dose, or cumulative dose of PGE1 and the incidence of side effects [18]. Among 435 neonates who received PGE_1_ treatment, Alghanem et al. reported that 175 (40%) had fever but only 1 (<1%) had culture-positive bacteremia [20]. Various adverse effects were found in our cohort of PGE1-treated infants, including respiratory (apnea and need for ventilation), cardiovascular (bradycardia, hypotension, and edema), and gastrointestinal. Our data revealed ventilated infants were more likely to receive early PGE1 treatment, and we found a trend towards increased ventilation in infants treated with higher maximal and cumulative doses of PGE1. Apneas are a well-documented short-term complication of PGE1 treatment, and may lead to a necessity for ventilation in affected infants. In our study, we found accumulating doses of PGE1 to be associated with gastrointestinal adverse effects and cessation of feeds in treated infants. Bone cortical proliferation and clinical periostitis associated with short- and long-term use of PGE1 have been reported as early as within 9 days of initiation of treatment in newborns, and have been shown to completely resolve following cessation of treatment [21]. We did not find infants exhibiting these phenomena in our study, although some of the infants were treated with PGE1 for up to 36 days.

In our cohort, none of the infants, including the minority who received prolonged PGE1 treatment, have developed clinical or radiological gastric obstruction. An additional study, including 22 infants who received long-term PGE1 infusion (mean duration of 38 days), reported adverse effects including fluid and electrolyte imbalance, gastric outlet obstruction, and feeding difficulties [22].

Both neonatologists and cardiologists are struggling with the dilemma of feeding in infants treated with PGE1_._ Feeding intolerance, gastric mucosal hypertrophy, and NEC were reported as associated with feeding newborns with ductal-dependent heart defects preoperatively, while receiving PGE1 treatment [23,24]. However, adequate nutrition is crucial for infants with congenital heart defects, as many infants are considered malnourished at the time of cardiac surgery, more so infants with cyanotic heart defects [25,26,27]. Malnutrition in these infants is multifactorial, and is due to both decreased intake, as nutritional practices have great variability between centers, and increased energy expenditure, due to their basic disease [28,29]. It has been demonstrated that malnutrition in critically ill infants has been associated with adverse outcomes, including increased mortality, and longer ventilation and hospital stay [30,31].

The incidence of NEC among term infants with congenital heart defects is 3–5% [32], and infants with congenital heart defects and NEC have an extremely high mortality rate (19–26%) [33]. The pathophysiology of NEC in infants with congenital heart defects is most probably multifactorial, including both acute and chronic circulatory changes [34]. These circulatory changes include the “diastolic steal” phenomena in duct-dependent infants, leading to impaired mesenteric blood flow [32]. Carlo et al. have demonstrated retrograde diastolic flow in 47% of infants with congenital heart defects who developed NEC compared to 15% in infants who did not. In this study, there was no association between ductal patency or reliance on ductal patency and the risk of developing NEC [35]. This observation is enforced by studies using splanchnic near spectrophotometry (NIRS) to demonstrate reduced regional oximetry in infants with congenital heart defects who have developed NEC [36,37]. An additional mechanism is episodic gut hypoperfusion leading to mesenteric ischemia and mucosal barrier injury [34]. Multiple studies have demonstrated that low cardiac output and need for inotropes is a risk factor for NEC among infants with congenital heart defects [32,38]. Other factors, such as abnormal abdominal vasculature [37] and gut-related factors including feed type, mucosal barrier, and gut microbiome, may play a part in the development of NEC in a full-term infant with a congenital heart defect [34].

In infants with duct-dependent lesions and PGE1 treatment, both the underlying disease and the treatment play a role in increasing the risk for NEC.

A recent retrospective study by Choi et al. reported 3% (10 infants) of 355 duct-dependent infants receiving PGE1 developer NEC. Of these, 20% developed surgical NEC (stage IIIB) and 20% had severe NEC (stage IIIa). They reported that duct-dependent pulmonary circulation, single ventricle physiology, and lower birth weight and gestational age were the main risk factors for developing NEC. In addition, they found that duct-dependent infants who developed NEC had an over fourfold risk of death compared to infants with duct-dependent heart defects who did not develop NEC [39]. This study did not examine the effects of PGE1 treatment characteristics on the risk to develop NEC. A study of 458 infants with congenital heart defects, 361 of then requiring PGE1 treatment, reported only 4 cases of surgical NEC, all of them in infants with additional risk factors—small for gestational age, near-term gestation, and additional congenital or chromosomal abnormalities. To note, 97% of the infants in this study were fed prior to cardiac surgery, the vast majority were on full enteral feeds, and all were fed at least 45 mL/kg/day [40]. Day et al. described necrotizing enterocolitis (NEC) in 18 of 177 newborns treated with PGE1 for duct-dependent congenital heart defects. The authors reported they did not find an association between enteral feeding of newborns who received PGE1 treatment and development of NEC [41]. A large retrospective study of 6710 newborn infants with duct-dependent congenital heart defects and PGE1 treatment by Becker et al. reported very low risk (0.3%) of developing NEC, with the risk factors being prematurity and single ventricle physiology. Although they found a trend towards increased incidence of NEC per infant days in enterally fed infants, this finding was far from being statistically significant [42].

Our data did not demonstrate an association with NEC or NEC associated radiological findings such as portal air or pneumatosis intestinalis on abdominal radiographs with PGE1 treatment characteristics. This finding agrees with previous publications [40,41].

In our NICU, which serves as a referral center for congenital heart defects from the greater Jerusalem district, we routinely feed infants receiving PGE1 treatment. As most infants are taken to surgery within a few days, we seldom encounter long-term gastrointestinal adverse effects such as stomach hypertrophy. However, some infants develop feeding intolerance and NEC while receiving PGE1 treatment. We found cumulative PGE1 doses to be associated with the frequency of gastrointestinal adverse effects. However, this association was not found for NEC. NEC is the leading gastrointestinal cause of death in the NICU, resulting in not only increased mortality but also long-term gastrointestinal and neurodevelopmental complications [43]. Clearly, these results should be further considered when treating preterm infants or those small for gestational age with PGE1. In an international cohort study that included 609 infants born at 24 to 31 weeks’ gestation with birth weights < 1500 g, it has been shown that significantly preterm infants with isolated, non-chromosomal, severe congenital heart defects exhibited two to three times higher odds for in-hospital mortality than infants without CHD. Moreover, while mortality declined with each week of increasing gestational age in infants without congenital heart defects, it remained high in infants with congenital heart defects, irrespective of gestational age [44].

Other, less significant adverse effects such as tachycardia and edema were associated with maximal dose and duration of treatment with PGE1, respectively.

Multiple studies reported a variety of gastrointestinal adverse effects of PGE1 treatment, besides NEC, more so in prolonged treatment with increased cumulative doses. Cucerea et al. reported over 25% of feeding intolerance in neonates receiving PGE1 treatment [45]. This observation is reinforced by our data and by additional studies [22]. Furthermore, PGE_1_ treatment and higher cumulative doses were reported as associated with gastric antral mucosal hyperplasia [46,47]. Nevertheless, the practice of feeding neonates while receiving PGE1 treatment and awaiting surgery is standard, and is supported by studies showing shorter length of stay in infants that were fed preoperatively [48].

In most centers worldwide, the initial dose of PGE1 is 0.05–0.1 mcg/kg/min, a regimen that has not changed since the introduction of PGE1 treatment in the mid-seventies of the 20th century [49]. Recent studies report much lower doses, of 0.01–0.02 mcg/kg/min, may achieve the same results while reducing adverse effects [12]. Haughey et al. developed a standardized PGE1 administration protocol for a defined group of prenatally diagnosed and hemodynamically stable infants with ductal-dependent disease. Their study demonstrated that initiating PGE1 treatment in this group using a dose of 0.01 mcg/kg/min has successfully maintained ductal patency, while reducing short-term adverse effects. They reported significantly less hyperthermia and a trend toward less apnea in the low-dose group. In addition, the low-dose group did not require more dose adjustments or emergency interventions than the higher-dose group (0.03–0.05 mcg/kg/min) [50]. Another study reported very low doses of PGE1 (0.003–0.05 mcg/kg/min) were sufficient for maintaining duct patency. Their data suggested that higher doses may be required for infants with duct-dependent systemic flow [51]. A recent study by the same investigators demonstrated that a dose of 0.01 mcg/kg/min of PGE1 was efficient in maintaining ductal patency in 83% of infants, and there were reduced adverse effects [13].

In vitro studies of long-term lipo-PGE1 administration reported little intimal cushion formation, EP4, the ductus arteriosus dominant PGE_2_-receptor, downregulation, and elastic characteristics of the ductus arteriosus. These results suggest the dosage of PGE1 may be decreased after a definite administration period [52].

Our data did not indicate an association between PGE1 maximal dose and gastrointestinal adverse effects; however, we did find that higher cumulative doses were significantly associated with a higher rate of adverse effects. Therefore, one may conclude that lower PGE1 doses may allow for enteral feeding even in infants that require prolonged treatment while awaiting surgery.

Our study is limited by its retrospective nature, as data in medical records may be incomplete. However, we have analyzed a group of infants with duct-dependent lesions treated in a single center with uniform practice and guidelines.

Most infants in resource-rich countries do not undergo a lengthy wait for surgery while receiving PGE1 treatment. However, some infants are born with low birth weights; early gestation; or other reasons for delayed surgical treatment.

Our data suggest that enteral feeding is safe in infants with ductal-dependent lesions receiving PGE1 and is not associated with NEC. However, in these infants, cumulative PGE1 dose may have a significant effect on gastrointestinal adverse effects; therefore, lower PGE1 doses should be considered.

## Figures and Tables

**Table 1 children-10-01572-t001:** Population characteristics and PGE1 dose.

	Median ± SD	Range
Gestational age (weeks)	39 ± 1.4	(36–42)
Birth weight (grams)	3122 ± 476.09	(2050–4130)
Age at surgery (days)	6 ± 6.91	(0–39)
NICU stay	7 ± 8.57	(1–39)
Age at PGE1 initiation (days)	0	(0–7)
PGE1 dose (mcg/kg/min)	0.05 ± 0.05	(0.01–0.5)
PGE1 cumulative dose (mcg/kg)	1035.72 ± 1574.1	(99–10,327.97)
PGE1 maximal dose (mcg/kg/min)	0.05 ± 0.02	(0.03–0.1)
Duration of PGE1 treatment (days)	5 ± 6.78	(0.5–39)

**Table 2 children-10-01572-t002:** Characteristics of congenital heart defects.

		N (%)
Defect type		
	Right-sided obstructive disease *^1^	31 (37.8%)
	Left-sided obstructive disease *^2^	32 (39%)
	Other *^3^	19 (23.2)
Prenatal diagnosis		
	Yes	66 (80.5%)
	No	14 (17.1%)
	Missing information	2 (2.4%)

*^1^ defect characterized by right-sided obstructive disease, i.e., severe pulmonary blood flow restriction including pulmonary atresia, critical pulmonary stenosis, and tetralogy of Fallot with severe stenosis. *^2^ defects characterized by left-sided obstructive disease, i.e., severe restriction in systemic blood flow including aortic stenosis, coarctation of aorta, interrupted aortic arch, and left heart hypoplastic syndrome. *^3^ other: transposition of great arteries.

**Table 3 children-10-01572-t003:** Frequency of PGE1 adverse effects.

Adverse Effect		N (%)
Respiratory	Apneas	17 (20.7)
Cardiac	Bradycardia	3 (3.7)
	Tachycardia	3 (3.7)
	Arrhythmia	4 (4.9)
	Hypotension	3 (3.7)
	Anuria	1 (1.2)
	Flushing/rash	3 (3.7)
	Edema	15 (18.3)
Hyperthermia		15 (18.3)
Neurological	Jitteriness	1 (1.2)
	Seizures	1 (1.2)
Gastrointestinal	Feeding intolerance	2 (2.4)
	Necrotizing enterocolitis	1 (1.2)
	Peritonitis	1 (1.2)
	Pneumatosis intestinalis (ABXR)	1 (1.2)
	Total GI adverse effects	11 (13.4)
Other	Thrombocytopenia	14 (17.1)
	Hypoglycemia	9 (11)
	Hypomagnesemia	3 (3.7)
	Sepsis	7 (8.5)

## Data Availability

Data regarding this study may be provided upon request.

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
