# Peer review of "Cumulative Dose of Prostaglandin E1 Determines Gastrointestinal Adverse Effects in Term and Near-Term Neonates Awaiting Cardiac Surgery: A Retrospective Cohort Study"

_children, 2023, doi:10.3390/children10091572_

Round 1

Reviewer 1 Report

General Comments:

The manuscript aims to discuss the effects and associations of PGE1 treatment in infants with ductal-dependent congenital heart defects. The topic is of clinical importance and has the potential to contribute valuable insights to the field. However, there are several areas that require attention and clarification.

Major Comments:

  1.  

  2. Introduction & Background: The introduction should provide a more comprehensive background on the significance of PGE1 treatment in infants with ductal-dependent congenital heart defects. As in currently a general description is given, however need to compare with previous existing literature and highlight the aim of the study to fill the existing gap in literature

  3.  
  4. Methodology:

    •  
    • The criteria for patient selection should be clearly defined.
  5.  
  6. Discussion:

    •  
    • The association between PGE1 treatment variables and adverse effects needs further elaboration. Discuss the clinical implications of these findings and how they compare with existing literature.
  7. Conclusion: Would suggest a separate paragraph with heading for conclusion.

Author Response

The authors appreciate the comperhensive review of the manuscript and wish to thank the reviewer.

Major Comments:

Introduction & Background: The introduction should provide a more comprehensive background on the significance of PGE1 treatment in infants with ductal-dependent congenital heart defects. As in currently a general description is given, however need to compare with previous existing literature and highlight the aim of the study to fill the existing gap in literature

We thank the reviewer for this comment. We have added the aim of the study to the introduction. (lines 57-59)

3.Methodology:

The criteria for patient selection should be clearly defined.

The authors appreciate this important comment, we have clarified the inclusion and exclusion criteria. (lines 62-65)

4.Discussion:

The association between PGE1 treatment variables and adverse effects needs further elaboration. Discuss the clinical implications of these findings and how they compare with existing literature.

The authors wish to thank the reviewer for this comment. The discussuion has been elaborated as per the comment.

  1. Conclusion: Would suggest a separate paragraph with heading for conclusion.

The conclusion was moved to a separate paragraph as per the reviewers’ comment.

Reviewer 2 Report

General comment

The authors investigated a retrospective cohort study to evaluate the relationship between PGE1 usage and adverse effect in neonate patients with ductus-dependent congenital heart defects. They found that cumulative PGE1 dose is associated with gastrointestinal adverse effects in their population and concluded that lower doses of PGE1 should be considered in neonates expecting prolonged PGE1 treatment.

The reviewer thinks this study's concept is interesting and important because the appropriate usage of PGE-1 is a crucial matter in these patients. However, there are some concerns to be resolved in order to clarify and improve the content of the manuscript.

Specific comment

Major comment 1

The introduction of the manuscript does not include what is unknown about the usage of PGE1 to date and what the purpose of this study is, and these should be stated clearly.

Major comment 2

What are the exclusion criteria in this study? For example, did the authors include patients with chromosome anomalies? If so, that should be the limitation of this study because the patients with chromosome anomalies sometimes tend to have more adverse events which may be the bias of this study.

Major comment 3

In Table 4, there was a trend (p = 0.090) towards more ventilated infants as the maximal dose of PGE1 increased, however, the maximal PGE1 dose is completely the same between the patients with and without ventilation therapy (median 0.05, range 0.03-0.1). In addition, although the authors said that there was a trend towards more ventilated infants as accumulative doses of PGE1 increased, the p-value is 0.987, and therefore is no statistical difference. There should be some mistakes and the authors should correct them.

Major comment 4

In L125, what means the “value of 0.715”? Is it an area under the curve? In addition, what was the positive predictive value of the ROC curve? They should be added clearly in the manuscript. The ROC curve should be added as a Figure for better understanding.

Major comment 5

The third and fourth paragraphs in the Discussion section are only the review of past literature. The authors should discuss the association between the review and the result of the current study in these paragraphs.

Major comment 6

The sixth and seventh paragraphs of the Discussion part are too short and do not include a discussion according to the result of the current study. These should be modified appropriately. 

Major comment 7

Paragraphs about NEC are scattered and confusing (paragraphs 3,5,7, and 8 in the Discussion), and thus they should be combined into one or two paragraphs.

Minor comment 1

The title includes “term neonate”, however, this study includes a baby with a gestational week less than 37 weeks. Therefore, the reviewer thinks that the title does not accurately describe the current study.

Minor comment 2

The abbreviation “DIC”(L43) and “GERD”(L44) is unnecessary because they never appear in the manuscript again.

Minor comment 3

The reviewer thinks the PGE1 used in this study is not a lipo-PGE1. Lipo-PGE1 is considered safe to use at lower doses compared to PGE-1 and is frequently used in some countries. Please add in the text the status of the use of lipo-PGE1 in the authors’ country. Furthermore, if they were able to use lipo-PGE1, it should be noted as a limitation that the effect of lipo-PGE1 was not investigated in this study.

Minor comment 4

The definition of bradycardia(< 100 bpm) and tachycardia (> 180 bpm) is ambiguous because even untreated, healthy newborns can have these heart rates during crying and sleep.

Minor comment 5

The definition of hyperthermia (body temperature > 37.50C or 38?) in unclear and should be corrected.

Minor comment 6

In the primary outcome, what is the definition of “short-term” and “intermediate-term”?

Minor comment 7

In the statistical analysis, abbreviations of Mann-Whitney (M-W) and Kruskal-Wallis (K-W) are unnecessary because they never appear in the manuscript again.

Minor comment 8

In L54-56 and L98, the font style seems to be different from other parts.

Minor comment 9

Table 4 is difficult to read, because the positions of adverse outcome, Yes, No are not uniform, and the rows of values corresponding to “Yes” and “No” are misaligned.

Minor comment 10

In Table 4, the median value of cumulative PGE1 dose in patients with no ventilation should be incorrect, because the median value (6.2) is out of range (99-5454). 

Minor comment 11

In Figures 1 and 2, the p-value should be added.

Minor comment 12

In L124, “ROC” should be spelled out because it appears for the first time. 

Minor comment 13

In L128-129, the authors said that hyperthermia and apneas were associated with maximal dose of PGE1, however, the p-values are 0.052 > 0.05 and thus they are statistically incorrect descriptions. This sentence should be modified appropriately.

Minor comment 14

The first paragraph in the Discussion section is a background of this study rather than a discussion and should be deleted or moved to the Introduction part.

Minor comment 15

The last sentence of the fifth paragraph in the Discussion part (L174-176) is not derived from the content within the paragraph or the result of this study, because the current study did not investigate or discuss the association of ventilation therapy and apnea. The authors should modify this sentence appropriately.

Minor comment 16

The last paragraph of the manuscript includes too many semicolons and thus is difficult to read. 

Minor comment 17

The authors said “Our data suggest that enteral feeding is safe in these infants” in L210-211, however, the meaning of this sentence is unclear. What the “these infants” mean? Is it an infant with a shorter waiting time for surgery, or an infant with low birth weights, early gestation, or other reasons for delayed surgical treatment?

Reviewer 3 Report

I read with interest the article by Shlomai and colleagues “ Cumulative dose of Prostaglandin E1 determines gastrointestinal adverse effects in term neonates awaiting cardiac surgery: a retrospective cohort study

Among the remarks and commentary that I would submit to the authors,

Though the subject is of interest, this retrospective analysis of 82 neonates receiving PGE-1 IV as part of a preoperative management strategy is pleagued by all the confounding factors that the authors failed to integrate into their data collection and analysis.

As the sickest patients (for those in whom a ductal closure might be rapidly fatal), or the one presenting at the latest stage (where ductal reopening if of utmost importance) will likely receive the highest doses to start with, this will finally results in the cumulative dose provided.

Most of the morbidity that the authors relates to the PGE1 can also be the morbidity linked to the severity of the clinical status of those neonates, so that the respective role of each of those variables is uncertain.

The authors could probably strengthen the manuscript by stratifying their cohort according to the available severity scores at each of the time point chosen (initiation of PGE1, maintenance of PGE1 and just prior to surgical repair, for example). The PRISM score or the P-SOFA score are readily available, even in a retrospective analysis.

The correlation between morbidity, clinical severity score and PGE1 doses could then be analysed in a more granular way.

Secondly, the data can be confusing :

-In table 1, the PGE1 cumulative dose is provided in mg/kg, but when one accounts for 0.05 mcg/kg.min, for a 3.1 Kg neonate receiving a 5-day course of treatment, the calculation is therefore

0.05 * 1440 * 5, which provides the cumulative dose received (per Kg) = 360 mcg total … can the authors reconcile the discrepancy noticed ?

-Hyperthermia should be defined as a single value, which in most neonatal centers, will be set at 38°Celsius.

Minnor improvement  requested

Author Response

Though the subject is of interest, this retrospective analysis of 82 neonates receiving PGE-1 IV as part of a preoperative management strategy is pleagued by all the confounding factors that the authors failed to integrate into their data collection and analysis.

As the sickest patients (for those in whom a ductal closure might be rapidly fatal), or the one presenting at the latest stage (where ductal reopening if of utmost importance) will likely receive the highest doses to start with, this will finally results in the cumulative dose provided.

Most of the morbidity that the authors relates to the PGE1 can also be the morbidity linked to the severity of the clinical status of those neonates, so that the respective role of each of those variables is uncertain.

The authors could probably strengthen the manuscript by stratifying their cohort according to the available severity scores at each of the time point chosen (initiation of PGE1, maintenance of PGE1 and just prior to surgical repair, for example). The PRISM score or the P-SOFA score are readily available, even in a retrospective analysis.

The correlation between morbidity, clinical severity score and PGE1 doses could then be analysed in a more granular way.

The authors wish to thank the reviewers’ comments.

As this is a retrospective study, the authors cannot answer some of the queries the reviewer had. However, as most infants have begun PGE1 treatment on the first day of life, and infants who didn’t survive 96 hours or had chromosomal or other major anomalies were excluded, the authors feel that severe outcomes were not associated with patient characteristic in a manner that would be likely to alter the results. Secondly, we have examined defect time, and did not find that this had influenced our results.

Secondly, the data can be confusing :

-In table 1, the PGE1 cumulative dose is provided in mg/kg, but when one accounts for 0.05 mcg/kg.min, for a 3.1 Kg neonate receiving a 5-day course of treatment, the calculation is therefore 0.05 * 1440 * 5, which provides the cumulative dose received (per Kg) = 360 mcg total … can the authors reconcile the discrepancy noticed ?

-Hyperthermia should be defined as a single value, which in most neonatal centers, will be set at 38°Celsius.

The authors appreciate the reviewers’ comments.

  1. The discrepancy in table 1 has been clarified as per the reviewers’ comment.
  2. Hyperthermia was defined as body temperature > 37.5oC

Round 2

Reviewer 2 Report

The authors corrected their manuscript appropriately according to the reviewer's suggestion. There are no further comments.